# High-Output Lotus-Leaf-Bionic Triboelectric Nanogenerators Based on 2D MXene for Health Monitoring of Human Feet

**DOI:** 10.3390/nano12183217

**Published:** 2022-09-16

**Authors:** Like Wang, Huichen Xu, Fengchang Huang, Xiaoma Tao, Yifang Ouyang, Yulu Zhou, Xiaoming Mo

**Affiliations:** Center on Nanoenergy Research, Guangxi Key Laboratory for Relativistic Astrophysics, School of Physical Science and Technology, Guangxi University, Nanning 530004, China

**Keywords:** triboelectric nanogenerator, MXene, graphene, energy harvesting, health monitoring

## Abstract

As versatile energy harvesters, triboelectric nanogenerators (TENGs) have attracted considerable attention in developing portable and self-powered energy suppliers. The question of how to improve the output power of TENGs using cost-effective means is still under vigorous investigation. In this paper, high-output TENGs were successfully produced by using a simple and low-cost lotus-leaf-bionic (LLB) method. Well-distributed microstructures were fabricated via the LLB method on the surface of a polydimethylsiloxane (PDMS) negative triboelectric layer. 2D MXene (Ti_3_C_2_T_x_) and graphene were doped into the structured PDMS to evaluate their effects on the performance of TENG. Owing to merits of the MXene doping and microstructures on the PDMS surface, the output power of MXene-doped LLB TENGs reached as high as 104.87 W/m^2^, which was about 10 times higher than that of graphene-doped devices. The MXene-doped LLB TENGs can be used as humidity sensors, with a sensitivity of 4.4 V per RH%. In addition, the MXene-doped LLB TENGs were also sensitive to human body motions; hence, a foot health monitoring system constructed by the MXene-doped LLB TENGs was successfully demonstrated. The results in this work introduce a way to produce cost-effective TENGs using bionic means and suggest the promising applications of TENGs in the smart monitoring system of human health.

## 1. Introduction

With the development of various electronic devices toward miniaturization and portability, there is a growing demand for developing novel miniaturized and portable energy suppliers that can collect the low-frequency motions to drive the electronic devices [1]. As nascent and hotly developing energy harvesters, triboelectric nanogenerators (TENGs) can convert various forms of mechanical energy into electrical energy to drive miniaturized electronic devices. On the one hand, compared with electromagnetic generators used widely today, TENGs have many outstanding advantages such as low cost, flexibility, portability, low frequency operation, high voltage output, considerable energy conversion efficiency, and significant versatility [2,3,4]. On the other hand, TENGs also have advantages over other emerging energy-harvesting piezoelectric generators (PGs), especially under the ultra-low frequency working environment of less than 1 Hz [5]. The electrical energy can be harvested by using difference of charge attraction between the positive and negative triboelectric layers of TENGs; the greater the difference, the higher the output. For TENGs, the negative triboelectric layer plays a pivotal role in the efficiency of charge production and exchange. Commonly used negative triboelectric layer materials include polytetrafluoroethylene (PTFE) [6], fluorinated ethylene propylene (FEP) [7], polyvinylidene fluoride (PVDF) [8,9], and polydimethylsiloxane (PDMS). Among them, the PDMS polymer stands out from the numerous negative triboelectric layer materials due to its flexibility, good electronegativity, and easy processing to mass production. It is widely used in TENGs, self-powered sensors, and biosensing applications for energy harvesting [10,11,12]. It has been previously demonstrated that PDMS-based PGs and TENGs are used not only as wireless energy harvesters, but also as highly sensitive and flexible biocompatible pressure/strain sensors to detect vibration or human motion [13,14,15]. However, as an insulating organic polymer, PDMS prevents TENGs from generating a greater potential difference and higher output current when it is used as the negative triboelectric layer [16,17,18]. There are mainly two sorts of solutions with respect to this issue: one is chemical modification by adding functional groups to the PDMS polymer via chemical reactions [19,20]; the other is physical modification by doping non-reactive nanomaterials into PDMS (graphene [21], carbon [22], carbon nanotubes [23], or hexagonal boron nitride [24], etc.) or by constructing microstructures on the surface of the PDMS films [25,26]. Within the solutions, though constructing microstructures on the surface of PDMS improves the performance of TENG effectively, it is very costly to use photolithography techniques to produce molds for microstructure constructions. In this regard, producing microstructures on the PDMS surface in an easy and cost-effective way is highly desired.

In recent years, two-dimensional (2D) materials such as graphene, 2D-antimonene sheets [27], 2D transition metal dichalcogenides (TMDs) and others [28] have been widely used in the development and research of TENGs, owing to the particular properties induced from their unique structures. 2D MXenes that are based on transition metal carbide/nitride (Ti_3_C_2_T_x_ or Ti_4_N_3_T_x_, where T_x_ represents the surface functional groups like –O, –OH, and –F), have received considerable attention in producing high-performance TENGs due to their unique structures and extraordinary electrical properties [29,30,31]. The outstanding electrical conductivity of metallic MXenes can reduce the resistance of the composite film markedly. The introduction of MXenes into the composite film also increases the specific surface area, thus improving the effective contacting area of the TENGs. Moreover, the large number of -F groups generated on the surface of MXenes during the preparation process have strong electron adsorption ability. Therefore, modification of the PDMS by doping MXenes can improve the electron adsorption ability of the negative triboelectric layer of PDMS, and thereby enhance the output performance of TENGs [32,33]. Up to date, many techniques have been developed to use MXenes to produce high-performance TENGs, such as electrospun, spraying, liquid electrode, and 3D aerogels. Considering the synergy of microstructures and modification with MXenes into PDMS, the reported techniques might be relatively complex and thus unsatisfactory. Therefore, developing a simple and easy way is more favorable. In addition, as the preparation technology matures and volume production of MXene increases, the price of MXene shall become lower and lower. Graphene, as another typical 2D material with high electron mobility, low cost, and high yield, is also frequently used to modify the negative triboelectric layer of TENGs [28,34,35]. However, for PDMS modification, comparative studies on the advantages and disadvantages of both MXene and graphene are still lacking and more investigations are required.

Here, high-output TENGs are demonstrated by using a low-cost lotus-leaf-bionic (LLB) method, rather than photolithography, to produce well-distributed microstructures on the surface of the PDMS negative triboelectric layer. The performance of the TENGs is investigated by comparing the modifications of the PDMS film by doping 2D MXene (Ti_3_C_2_T_x_) and graphene. The maximum output power density of MXene-doped LLB TENG with optimal MXene mass ratio is found to be 104.87 W/m^2^, reaching about 10 times higher than that of the graphene-doped LLB TENG. The as-produced MXene-doped LLB TENG can be used as a humidity sensor with a sensitivity of 4.4 V per RH% and is able to drive low-power electronic devices, including electronic watches and green LED arrays. In addition, the MXene-doped LLB TENG can also derive energy from different human motions. and hence, a health monitoring system constructed by the MXene-doped LLB TENG for human feet is demonstrated for the first time in this work. The results in this work introduce a way of producing low-cost TENGs by bionic means and suggest the promising applications of TENGs in the smart monitoring system of human health.

## 2. Materials and Methods

**Materials.** Polyvinyl alcohol (PVA) was purchased from Sinopharm Chemical Reagent Co., Ltd. (Shanghai, China). PDMS and its curing agent were purchased from the DOW Chemical Company (DOWSIL SYLGARD 184, Midland, MI, USA) and used as the negative triboelectric layer. Polyamide 6 (PA6) was purchased from Dongguan Yixuan Plastic Co., Ltd. (Dongguan, China). MXene with flake-stacking morphology was supplied from Shandong Xiyan New Material Technology Co., Ltd. (Jinan, China). (See Appendix A). The MXene clusters range from several micrometers to tens of micrometers in size with a single-flake thickness of ~220 nm (Appendix A). Graphene powder with ultra-thin layers was purchased from Shenzhen Yuechuang Technology Co., Ltd. (Shenzhen China). (see Appendix A). Each graphene cluster ranges from a few micrometers to dozen of micrometers in size, and includes one or several layered graphene thin layers (Appendix A). All the materials were used as received, without further purification.

**Fabrication of structured PVA mold based on lotus leaf.** As shown in Figure 1a, flat and perfect lotus leaves were carefully chosen as the raw materials to prepare the PVA mold. After being washed with a tiny brush and rinsed with considerable amount of water, the lotus leaves were flattened, cut into circles with diameters of ~15 cm, and dried at room temperature. A total of 5 g of PVA particles was added to 45 mL of ultrapure water, and the solution was heated to 90 °C for 4 h with continuous magnetic stirring to ensure the complete dissolution of the PVA. The PVA solution was then poured on the surface of the trimmed lotus leaves; finally, the structured PVA molds that had inverse micro structures with respect to the lotus leaves were obtained after curing for 48 h at room temperature.

**Fabrication of structured MXene-doped PDMS film.** A range of MXene-doped PDMS mixtures (with MXene mass ratios to be 0%, 5%, 10%, and 15%) were configured to evaluate the impact on device performance. The MXene-doped PDMS mixtures were prepared by adding the MXene into the PDMS with magnetic stirring for 30 min at room temperature. Then, curing agent was added into the MXene-doped PDMS mixtures at a mass ratio of 10: 1 and stirred for another 30 min. The MXene-doped PDMS mixtures were spun onto the PVA mold at a speed of 500 rpm for 30 s. The PVA mold coated with the MXene-doped PDMS mixtures was put into an oven to cure for 2 h at 90 °C. The cured MXene-doped PDMS mixtures were peeled off from the PVA mold, and a series of structured MXene-doped PDMS films that had the same microstructures as the lotus leaves were obtained for use as the negative triboelectric layer. A series of graphene-doped PDMS films were introduced as a comparison with the MXene. The procedures of the graphene-doped PDMS films were similar to those of the MXene-doped PDMS films, except for the mass ratios of the graphene in the mixtures, which were 0.2%, 0.4%, 0.6% and 0.8%.

**Preparation of TENGs.** The lotus-leaf-bionic (LLB) TENG consisted of two individual parts: the upper part and the lower part, as shown in Figure 2a. A layer of PA6 film covered with copper electrode was fixed to an acrylic sheet to form the upper part of the LLB TENG. For the lower part of the LLB TENG, a layer of structured MXene-doped PDMS (or graphene-doped PDMS) film with copper electrode was fixed to another acrylic sheet. The size of each part was controlled to be 2 cm × 2 cm. Wires were firmly attached to every electrode for all the performance tests.

**Characterizations and measurements.** Morphologies of the structured MXene/PDMS (or Graphene/PDMS) films were obtained by using scanning electron microscopy (SEM, ZEISS Sigma-500). The X-ray diffraction (XRD) patterns were conducted on Rigaku smartlab3KW. The LLB TENGs were driven by a linear motor (Linmot E1100-COHC) to achieve periodic motions during electrical measurements. The voltage and current were tested by an oscilloscope (Tekteonix MD03012, Beaverton, OR, USA). To obtain more accurate measurement data, a low-noise current preamplifier was used (SRS, SR570). The short-circuit charge was measured by an electrometer (Keithley 6514, Cleveland, OH, USA). The change in potential distribution was simulated by COMSOL software (COMSOL Multiphysics 5.5, Stockholm, Sweden). Surface roughness and Kelvin probe force microscopy (KPFM) analysis were measured using atomic force microscopy (AFM, MFP-3D-SA).

## 3. Results and Discussion

Figure 1a briefly shows the fabrication process of structured MXene-doped PDMS film via bionic means that uses lotus leaf as the template to produce molds. Since MXene-doped PDMS film is prone to fractures when it is peeled off from the lotus leaf directly, we firstly peel off a PVA film from the lotus leaf as a reverse mold to copy the inverse microstructures. Another peeling-off process is then conducted on the reverse mold of PVA with PDMS to obtain structured MXene (or graphene)-doped PDMS film that has the same microstructures as the lotus leaf. A spin-coating process is performed on the PVA reverse mold to get the structured MXene (or graphene)-doped PDMS film with controlled thickness. Finally, the as-obtained MXene (or graphene)-doped PDMS films are cut into the same size (e.g., 2 cm × 2 cm) as the negative triboelectric layers to further construct a vertical contact-separation mode TENG (together with the same size of PA6 film as the positive triboelectric layers). It should be noted that the triboelectric layers are rationally selected according to the ranking of the triboelectric series of materials in the literature [36].

Figure 1b shows the photographs of the as-prepared MXene-doped PDMS films (2 cm × 2 cm) with different MXene mass ratios. The cuttable squared morphologies of the as-prepared MXene-doped PDMS films confirm the feasibility of mass production of the MXene-doped PDMS films via the LLB method. Figure 1c depicts the SEM image of the surface morphology of the as-prepared MXene-doped PDMS film (with 10 wt.% MXene), from which it can be clearly observed that frustum-shaped microstructures are uniformly formed and distributed throughout the surface of the MXene-doped PDMS film, which is beneficial to improving the contact area to produce more charge for the as-prepared LLB TENG. It should be noted that doping the MXene (or graphene) into PDMS has little influence on the structured morphology of the PDMS surface (see Appendix A). After the MXene-doped PDMS film is cured, the status of the MXene in the MXene-doped PDMS composite film is investigated via XRD. As depicted in the XRD results in Figure 1d, a diffraction peak related to PDMS at ~12° decreases substantially with increasing the mass ratio of MXene. The decrease in the intensity of the crystal peak represents the conversion of the crystalline domain in the PDMS to the amorphous domain. Besides, another broadband diffraction peak related to PDMS at ~22° emerges with increasing the mass ratio of MXene, indicating that doping MXene in the PDMS can influence the crystalline process to some extent, but without dominating the crystallinity of the whole polymer host. Disappearance of the diffraction peak related to MXene at ~9° confirms that MXene is not dominant in PDMS and is well-distributed in the PDMS host. Similar results can also be obtained for the graphene-doped PDMS composite films from the SEM and XRD measurements (see Appendix A). To more clearly present the distribution of MXene and graphene in PDMS, cross-sectional SEM images of MXene (or graphene)-doped PDMS film are also shown in Appendix A, confirming the well distribution of MXene and graphene in the PDMS film, which is consistent with the XRD results. To form this conclusion more clearly, Raman spectra for the MXene- and graphene-doped PDMS composite films are also tested (see Appendix A). It is evident that the intensity of the original Raman peaks of the PDMS decreases with the increase of the mass ratio of MXene (or graphene), but there is no change in the wavenumber position. This result suggests that the doped MXene (or graphene) material is well-dispersed and well-preserved in the PDMS polymer, in accordance with the XRD results. It should be mentioned that the bending mode of water at 1625 cm^−1^ can also be observed in the Raman spectra of the MXene- and graphene-doped PDMS films, which might be introduced in the preparation process of the PVA molds. The presence of water could cause changes in the conductivity of the composite films [37,38] and influence the output performance of TENG. These results demonstrate that the MXene- and graphene-doped PDMS composite films with specific microstructures are able to be produced effectively by the LLB method, having little influence on the crystal structure of the MXene or graphene in the PDMS host.

The operating principle of the LLB TENG is based on the triboelectrification effect between the positive (PA6) and negative (MXene- or graphene-doped PDMS) triboelectric layers as they are vertically contacted and separated periodically, as shown in Figure 2a. In the initial state, the two parts of the device remain separated, and the upper and lower triboelectric layers are electrically neutral to each other. When an external force is provided for extrusion, the MXene-doped PDMS (or graphene-doped PDMS) negative triboelectric layer shall gradually make contact with the PA6 positive triboelectric layer. According to the triboelectric effect and the triboelectric series ranks, the contact between the MXene-doped PDMS (or graphene-doped PDMS) layer and the PA6 layer transfers equal but opposite charges. During the process of triboelectrification, electrons are transferred from the PA6 layer to the MXene-doped PDMS (or graphene-doped PDMS) layer, which results in a positive charge on the plane of the PA6 layer and a negative charge on the plane of the MXene-doped PDMS (or graphene-doped PDMS) layer, as shown in Stage I in Figure 2a. When the applied external force is gradually reduced, the elastic nature of the materials shall separate the PA6 and MXene-doped PDMS (or graphene-doped PDMS) triboelectric layers. Once the two surfaces are separated, a potential difference can be generated to drive the induced electrons to flow from the underside copper electrode to the top copper electrode through an external load to balance the potential difference, as shown in Stage II in Figure 2a. When the potential difference reaches equilibrium, the movements of the induced electrons also reach equilibrium so that the two copper electrodes connected by an external load have equal amounts of charges but with contrary polarities, as demonstrated in Stage III in Figure 2a. When the external force is applied again, the equivalent condition is broken again and an electron backflow from the top copper electrode to the underside copper electrode occurs, as shown in Stage IV in Figure 2a. Repeating the above process periodically, one can produce AC output continuously to the external circuit to drive the loads. Figure 2b shows the voltage signal of the LLB TENG during one working cycle, demonstrating the working status for each stage as illustrated in Figure 2a. The change in electrical potential distribution over one working cycle is simulated using COMSOL software, conforming the working principle of the LLB TENG, as shown in Figure 2c.

Surface properties of the triboelectric layers are analyzed by AFM and KPFM. The analysis results of surface roughness and surface charge potential are shown in Appendix A. As obtained from Appendix A, the Ra value of the MXene-doped PDMS is 91.9 nm, while the Ra value of the graphene-doped PDMS is 129.8 nm. The average surface roughnesses of both the MXene- and Graphene-doped PDMS films do not differ much from each other, though large fluctuations are observed on both films. Measurements of the surface charge potential show that the higher surface roughness of the graphene-doped PDMS renders the distribution of surface charge potential more uneven, as in Appendix A. The potential of MXene-doped PDMS varies from −7.7 V to −8.1 V, but it varies from −6.0 V to −7.0 V for the graphene-doped PDMS, which indicates that the potential gain of the MXene-doped PDMS negative triboelectric layer is greater than that of the graphene doping, and hence, TENG with the MXene-doped PDMS may have better output properties. The flat PA6 surface (with an Ra value of 10.41 nm) and a positive charge potential suggest that PA6 is of good quality to be used as the positive triboelectric layer, as shown in Appendix A.

To investigate the effect of MXene net content on the output performance of the LLB TENG, MXene-doped PDMS films with different MXene mass ratios (ranging from 0–15 wt.%) are prepared as the negative triboelectric layer. In comparison with the LLB TENG using pure PDMS film (0 wt.%), the open-circuit voltage, short-circuit current, and transfer charge of the MXene-doped LLB TENG are all significantly improved, as shown in Figure 3a–c. When the MXene content reaches 10 wt.%, the output efficiency of the LLB TENG reaches its maximum. At this time, the peak open-circuit voltage, short-circuit current, and transfer charge are 288 V, 20 μA, and 30.1 nC, respectively, which are 1.71, 2.98, and 1.30 times as those of the LLB TENG with pure PDMS film (168 V, 6.72 μA, and 24 nC, respectively). Note that when the mass ratio of the MXene exceeds 10 wt.% (e.g., 15 wt.%), the output performance of the LLB TENG reduces, indicating that doping more MXene is not always beneficial for the performance improvement of the MXene-doped LLB TENG. In fact, the open-circuit voltage of a TENG is positively correlated with the surface charge density of the triboelectric layers and the distance between the two triboelectric layers. The doping of MXene leads to an enhanced opportunity for the PDMS negative triboelectric layer to adsorb more charges due to the inclusion of more -OH, -F, and -O electron traps from MXene [28]. In this case, more charges shall be produced on the surface of the triboelectric layers with increasing MXene content, thus leading to a performance enhancement of the LLB TENG. Nevertheless, the addition of too many highly electronegative MXene materials will destroy the charge storage ability of the PDMS and increase the probability of electron overflow [39]. As a result, if the MXene is over-doped, the accumulated charges will spill over into the copper electrode, so that the electrical output performance of the LLB TENG with 15 wt.% or more MXene appears to be reduced.

To further evaluate the effect on the outputs of LLB TENG by doping 2D materials in the PDMS negative triboelectric layer, graphene, another typical 2D material, is used as the control experiment (see Appendix A). The preparation processes of the LLB TENG using graphene-doped PDMS are the same as those of the MXene-doped LLB TENG. The optimal doping mass ratio of the Graphene is investigated, and the results are depicted in Figure 3d–f, suggesting that the LLB TENG with 0.6 wt.% of graphene has the highest outputs. When graphene is added into PDMS, it can form electron traps similar to structures of microcapacitors, thus enhancing the TENG output with an increasing doping mass ratio. However, excessive addition of graphene will degrade the charge storage of the PDMS host for the same reason as the MXene, thus resulting in reduction of the output performance [40]. The peak open-circuit voltage, short-circuit current, and transfer charge are 264 V, 13.6 μA, and 31.7 nC, respectively. In the output parameters, all of them are much better than those of the pure PDMS-based LLB TENG. Nevertheless, in comparison with those of the MXene-doped TENG, the transfer charge is superior, which might be attributed to the fast electron mobility of graphene [41]. To further evaluate the difference in the output power for the MXene- and graphene-doped LLB TENGs, four devices of MXene-doped LLB TENGs (with 10 wt.% MXene in PDMS) and four devices of graphene-doped LLB TENGs (with 0.6 wt.% graphene in PDMS) are prepared. The output power of the LLB TENGs is calculated using the Equation [42]:$$P_{\mathrm{Density}}=I^{2}R/S$$
where *I* is the output current, *R* is the load resistance, and *S* is the contact area. The output power density and the respective current under different loads for the MXene- and graphene-doped LLB TENGs are recorded and shown in Figure 3g,h, respectively. For the MXene-doped LLB TENG, when the external load increases to 9 GΩ, the output power density reaches a maximum value as high as 104.87 W/m^2^ (with an average value of 87.01 W/m^2^), and the output current at this time is 2.16 μA, as shown in Figure 3g. For the graphene-doped LLB TENG, the maximum output power density is achieved to be merely 10.10 W/m^2^, as the external load is 9 GΩ (with average value of 8.85 W/m^2^) and the output current at that time is 0.67 μA, as shown in Figure 3h. The maximum output power for pure PDMS-based TENG is 4.85 W/m^2^ (with an average value of 4.55 W/m^2^), and the current is 0.46 μA, as shown in Appendix A. The above results reveal that 2D MXene-doped LLB TENGs have much better output power performance than the 2D graphene-doped LLB TENGs or the pure PDMS-based TENGs, which could be attributed to the permittivity difference between the structured MXene- and graphene-doped PDMS films and the higher potential gain of the MXene-doped PDMS film (See Appendix A). Compared with other MXene-based devices using different techniques in past years, the performance of LLB TENG has reached a very high level, as demonstrated in Table 1.

Owing to the outstanding performance of the MXene-doped LLB TENG, we are also interested in the effect of environmental relative humidity (RH) on the output performance of the device. The output voltage of the MXene-doped LLB TENG as a function of the RH value is investigated and presented in Figure 3i. A closed glove box and humidifier are used to control the RH value of the ambient environment. As the RH value increases from 20% to 80%, the output voltage decreases dramatically from 280 to 19.44 V, which fits a good linear reduction with sensitivity of 4.4 V per RH%. The output voltage of the MXene-doped LLB TENG is reduced because the moisture in the air ambient forms a thin water layer on the surface of the triboelectric layers, resulting in reduction of the surface electrostatic charge and hence decreasing the output voltage. The linear variation of the output voltage confirms the potential utilization of the MXene-doped LLB TENGs as sensitive humidity sensors.

Abilities of the MXene-doped LLB TENG in the applications of driving small electronic devices are further explored by using a rectifying circuit to convert the output alternating current to direct current, as shown in the inset of Figure 4a. The capacitor is used to smooth the output current and store the output energy to power the small electronic devices easily and quickly. Six different commercial capacitors, including 2.2, 4.7, 10, 33, 47, and 100 μF, are used to evaluate the charging efficiency of the MXene-doped LLB TENG. As displayed in Figure 4a, the charging rates (defined as the time rate of charging the capacitor to 2 V) are calculated to be 0.82679, 0.34238, 0.09524, 0.04387, 0.03665, and 0.01676 V/s for the above-mentioned six capacitors, respectively. To balance the charging efficiency and the capacity, the capacitor with capacitance of 33 μF is selected for further tests. As demonstrated in Figure 4b and Appendix A, when the MXene-doped LLB TENG is working, the 33 μF capacitor is charged and the voltage across the capacitor rises. Once the voltage reaches ~1.6 V, we stop the functioning of the MXene-doped LLB TENG and let the capacitor drive an electronic watch to display and support its normal working. As continuous discharging of the capacitor, the voltage across the capacitor decreases, but the display on the electronic watch can maintain for over 20 s. Once the display of the electronic watch disappears, we allow the MXene-doped LLB TENG function again, and the voltage across the capacitor rises back to ~1.6 V as the power continues to be supplied by the MXene-doped LLB TENG. The charging-discharging operations can be realized even after considerable cycles, indicative of the output stability of the MXene-doped LLB TENG. Figure 4c demonstrates use of the MXene-doped LLB TENG to directly light up 18 green LEDs (see Appendix A) without any other additional circuits. Notably, the MXene-doped LLB TENG also performs sensitively for capturing human body movements (including clapping hands, flapping arms, and up-and-down heel movements), as shown in Figure 4d–f.

The superior ductility and high sensitivity of the MXene-doped LLB TENG guarantee the utilization of the device in response to small pressure changes, especially the precise signals of the human body for health monitoring. In this regard, to further explore the application of the MXene-doped LLB TENGs in health monitoring, we design a health monitoring system for human feet by using 2 separate MXene-doped LLB TENGs. A facile way is provided to evaluate the healthy status (see Appendix A) of the feet by measuring the pressure changes of the two separate MXene-doped LLB TENGs. For demonstration purposes, the left foot is used as an example. The two MXene-doped LLB TENGs are fixed separately on an acrylic sheet about 4–5 cm from each other (see Appendix A) and denoted as inner or outer sensors. Three different results can be obtained from the variations of the detected output voltages for the three cases shown in Figure 5c,f,i. For the normal walking behavior of healthy feet, as shown in Figure 5a–c, one can observe histogram-like voltage signals received from the inner sensor and the outer sensor, indicating that the forces on the two separate MXene-doped LLB TENG sensors are balanced during walking of the healthy feet. The histogram-like shapes at the peak of the voltage signals from both sensors represents that the positive and negative triboelectric layers of each sensor have been in contact for a specific time (Δt) which is related to each person’s personal habits during walking. No spikes can be found in the voltage signals. Therefore, by evaluating the waveform of the voltage signals from both sensors, one may obtain the healthy information about the feet during walking. Reliability of the outer sensor under 10,000 working cycles clearly demonstrates the good stability and repeatability of the device (Appendix A). For the inversion case of sick feet, as depicted in Figure 5f, the force on the outer sensor is significantly stronger than the force on the inner sensor during walking. In comparison with the voltage signals from the healthy feet, the waveform appears to be very different. Very sharp spikes and deep valleys can be observed in the waveform from the inner sensor, as shown in Figure 5d,e. Such an inversion in one’s walking mode is harmful to one’s health and may result from some inborn diseases or poor walking habits. As a result, once such a waveform is detected, the monitored signals should be sent to the doctor and the walking mode (or feet) should be promptly corrected, especially for children between the ages of 3 to 12 years old. Finally, for the eversion case of sick feet, as depicted in Figure 5i, the force on the inner sensor shall be stronger than that on the outer sensor. As revealed from the output voltage signals in Figure 5g,h, multi-sharp spikes are also observed during walking in an eversion way. Thereby, if such a waveform is monitored when one is walking, he/she should be sent to the hospital to correct his/her feet or the walking habit. Note that the waveforms are very different for the three cases so that we can evaluate the possible disease of the human feet by simply evaluating the waveforms of output voltage signals. The above demonstration results show that the MXene-doped LLB TENGs can be used as sensitive monitoring sensors for the health of human feet.

## 4. Conclusions

In summary, fabrication of 2D MXene-doped high-output TENGs has been demonstrated by using the low-cost LLB method. Well-distributed microstructures are produced on the surface of the PDMS negative triboelectric layer via the LLB method. Addition of the 2D MXene to the PDMS film greatly improves the output performance of the LLB TENG. The output power density of the MXene-doped LLB TENG is as high as 104.87 W/m^2^, which is 10 times higher than that of the graphene-doped LLB devices. The MXene-doped LLB TENGs not only display extraordinary sensitivity to the environmental RH, but also present great potential in driving electronic devices (e.g., an electronic watch or an array of 18 green LEDs) and collecting energy from body movements. For the first time, the health monitoring system is successfully constructed to monitor the health of human feet by evaluating the voltage signals from the two separate MXene-doped LLB TENGs. The results of this work introduce a way to produce low-cost TENGs by bionic means, and demonstrate promising applications of the LLB TENGs in the smart monitoring system of human health.

## Figures and Tables

**Figure 1 nanomaterials-12-03217-f001:**
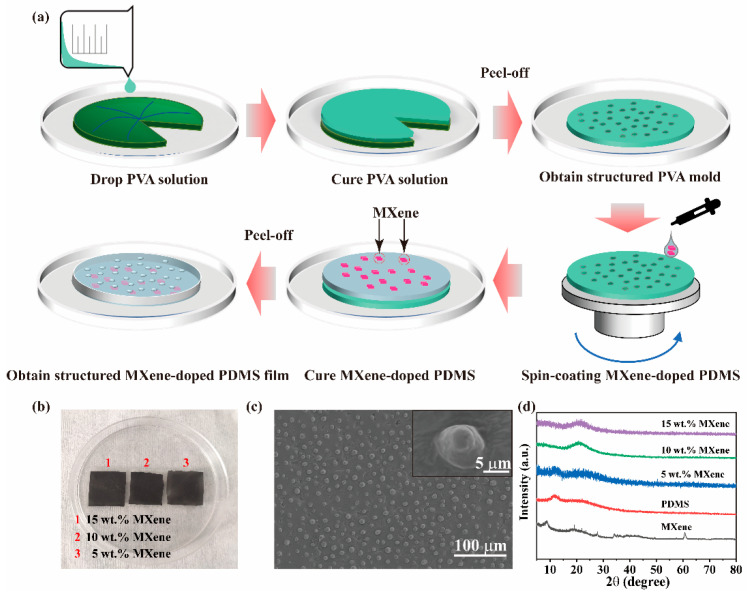
(**a**) Schematic diagram for preparation of structured MXene-doped PDMS film. (**b**) Photograph of the as-prepared MXene-doped PDMS films. (**c**) Typical SEM image of the MXene-doped PDMS film (with 10 wt.% MXene). (**d**) XRD patterns of the MXene-doped PDMS films in comparison with the pure PDMS and MXene.

**Figure 2 nanomaterials-12-03217-f002:**
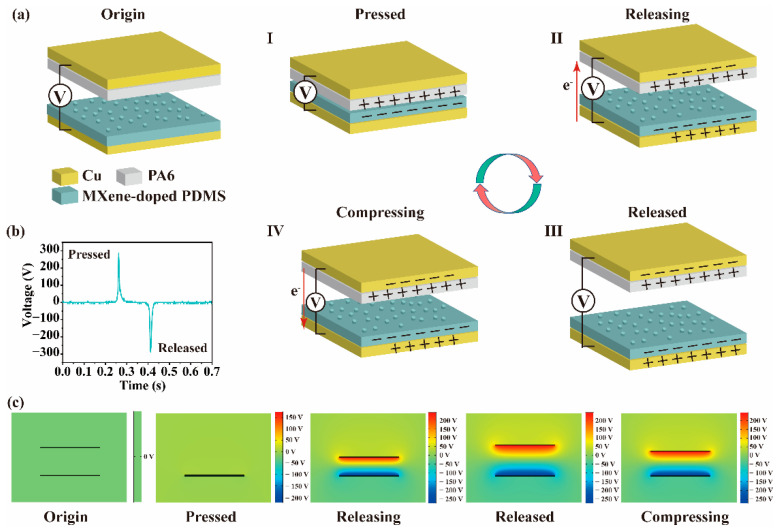
(**a**) Working principle of the LLB TENGs. (**b**) Voltage signal during one working cycle. (**c**) Simulation of potential distribution for one working cycle by COMSOL software.

**Figure 3 nanomaterials-12-03217-f003:**
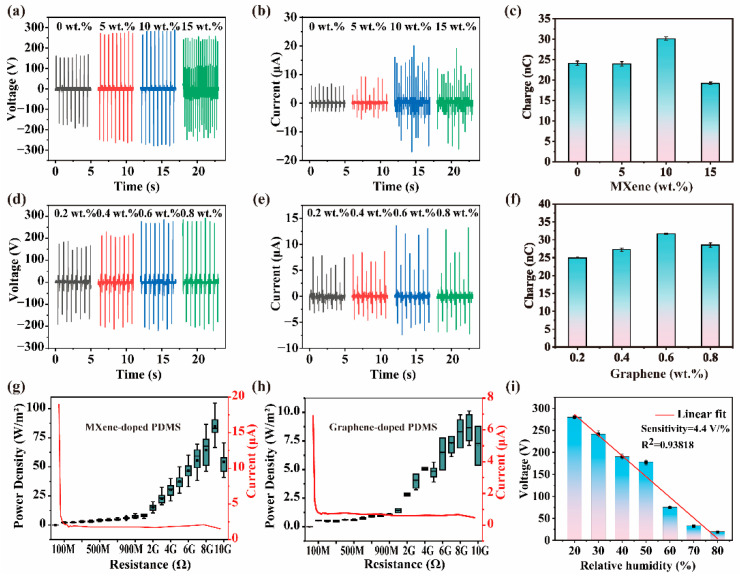
(**a**–**c**) Outputs for the LLB TENGs with varying the mass ratio of MXene in the structured MXene-doped PDMS film. (**d**–**f**) Outputs for the LLB TENGs with varying the mass ratio of graphene in the structured graphene-doped PDMS film. Power density and current curves under different load resistance for (**g**) MXene-doped PDMS film and (**h**) graphene-doped PDMS film. (**i**) Output voltage of the MXene-doped LLB TENG as a function of the RH of the ambient environment.

**Figure 4 nanomaterials-12-03217-f004:**
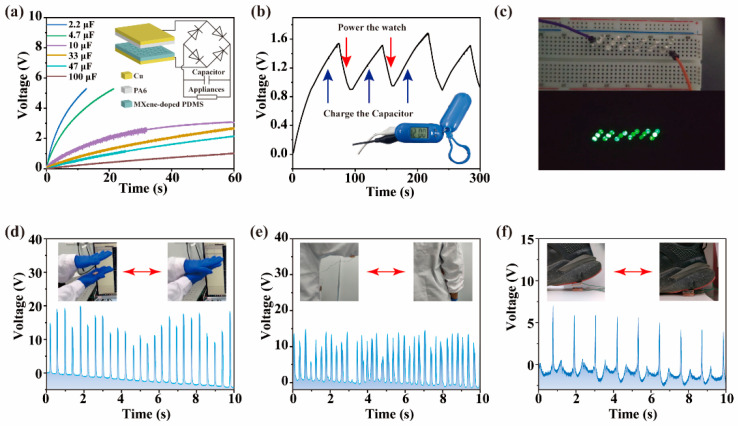
(**a**) Charging curves of the capacitors with different capacitances. The inset is the rectifier circuit using the TENG to drive the load. (**b**) Charging and discharging curve of the 33 μF capacitor when the MXene-doped LLB TENG is working to power an electronic watch (inset). (**c**) Utilization of the MXene-doped LLB TENG to drive 18 green LEDs. Sensing signals of the body movements with the MXene-doped LLB TENG induced by (**d**) clapping hands, (**e**) flapping arms, and (**f**) up-and-down heel movements.

**Figure 5 nanomaterials-12-03217-f005:**
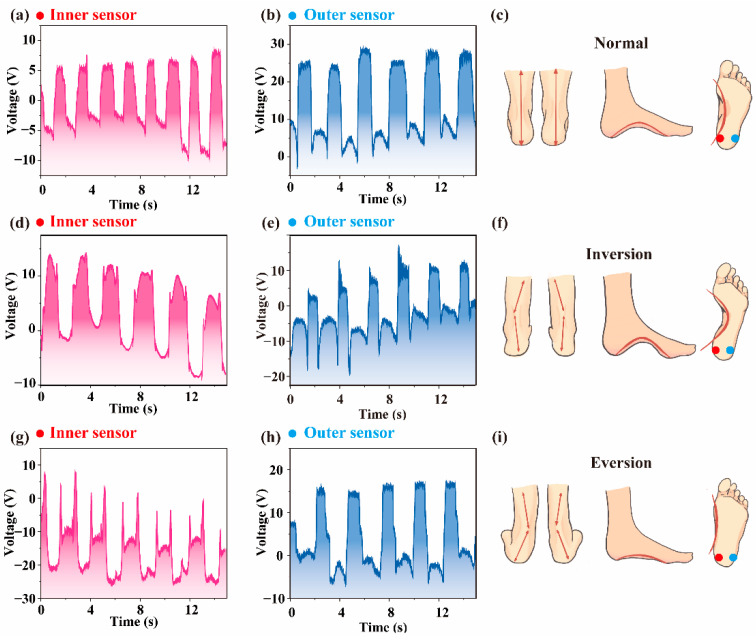
Voltage signals from the MXene-doped LLB TENGs for health monitoring of human feet: (**a**–**c**) normal, (**d**–**f**) inversion, and (**g**–**i**) eversion.

**Table 1 nanomaterials-12-03217-t001:** Performance comparison of the LLB TENG with other devices using MXene.

Materials	Handling Method	Size	Voltage	Current	Power	Ref.
PDMS/MXene-PA6	Lotus-leaf-bionic	2 × 2 cm^2^	288 V	20 μA	104.87 W/m^2^	This work
PVDF/MXene-Nylon 11	Electrospun layers	Φ 15 mm	270 V	40 mA/m^2^	4.02 W/m^2^	[43]
PET/AgNW@MXene	Electrospun layers	4 × 1.8 cm^2^	62 V	0.98 μA	-	[44]
PTFE/MXene-Cu	Spraying and CVD	5 × 5 cm^2^	397 V	21 μA	0.9 W/m^2^	[45]
CNFs/MXene-skin	Liquid Electrode	8 × 4 cm^2^	300 V	5.5 μA	504 mW/m^2^	[46]
CMCC/MXene-PVDF	3D aerogels	Φ 25 mm	115.3 V	0.78 μA	402.94 mW/m^2^	[47]
3D MXene-PDMS	3D aerogels	Φ 6.3 mm	45 V	0.6 μA	-	[48]

## Data Availability

The data is available on reasonable request from the corresponding author.

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
