# Peer review of "High-Output Lotus-Leaf-Bionic Triboelectric Nanogenerators Based on 2D MXene for Health Monitoring of Human Feet"

_nanomaterials, 2022, doi:10.3390/nano12183217_

Round 1

Reviewer 1 Report

     The authors have reported the usage of Mxene/PDMS composite for the triboelectric nanogenerator applications. After careful evaluation of the manuscript, I cannot recommend for publication at the current state as there are some queries  need to be addressed before publication in the Nanomaterials journal. The reason for my decision is as follows:

Comments:

1) The introduction section need to emphasize the usage of Mxene over other 2D materials reported and the novelty of the current work. Ref: 1. Nano Energy 77(2020) 105248; 2. Adv. Funct. Mater. 31 (2021) 2009994; 3. Mater. Adv., 1 (2020) 1644-1652

2) The surface potential charge of the triboelectric film is more essential in the fabrication of TENG and also for the selection of opposite triboelectric layer for effective output. Hence authors need to provide the KPFM analysis for the triboelectric layer in the manuscript.

3) The load matching data provided in Figure 3 (g-h) shows very high resistance of 8 to 9 Giga ohm which is not suitable for the practical application. This high resistance device is not applicable in the real time application and hence authors need to validate the practical applicability of the device.

4) The surface roughness of the triboelectric layer need to be measured and provided in the manuscript.

5) Why the authors chose graphene and Mxene in this work need to elaborately provided in the manuscript.

Reviewer 2 Report

The paper of L. Wang et.al represents a good application/engineering study of MXene based TENGs. The manuscript is well organized and structured, despite reporting some heterarchical data. While providing interesting new results, the paper lacks fundamental background and microstructural characterization of MXene films, necessary for clear understanding of the findings. The paper can be considered appropriate for Nanomaterials after the following issues are concerned.

1. Accounting for numerous recent reports on uses of MXenes for construction of triboelectric devices the novelty of the approach should be clearly stated and justified. The advantages of LLB structures over alternatives (electrospun layers, MXene liquid electrodes, 3D aerogels, etc) should be discussed. A direct comparison of the performance of the obtained TENGs (with LLB structure) over the alternatives in terms of technical feasibility and attained performance need to be provided in the discussion section. The resistance and performance of the device fabricated with no additives should also be reported.

2. Importantly, the paper lacks adequate microstructural characterization for MXene/PDMS (Graphene/PDMS) layers. The size of flakes of MXene and graphene is not reported. The distribution of MXene (graphene) flakes in PDMS layer remains unclear. The contacting area in triboelectronic device is not quantified. The authors are strongly advised to provide the above structural characterization of their samples (preferably with cross-section microscopy investigation).

3. The portions of Mxene and graphene additives to PDMS strongly differ. The authors need to provide a clear explanation on the nature of the strong difference of the resistance of the films (flake sizes, conductivity, etc). A comparison of the performance of films containing equal amount of nanoflake additives (in % atomic) also seems feasible.

4. Figure S4. is confusing. The authors provide MXene raman spectra for a limited spectral range (to <3200 cm-1), while illustrating full range spectra (to <6000 cm-1) for graphene containing samples. Representation of spectral region ~3500 cm-1 with specific water vibrations is necessary for MXenes. This becomes even more confusing with visible bending mode of water (~1685 cm-1) appearing with increasing MXene content in PDMS. As the quantity of absorbed water greatly influences TENGs performance, the specific water sorption characteristics and variation of interlayer spacing in MXene should be considered, or at least commented. The authors may refer to known resistance and sorption characteristics of MXene layers [https://pubs.acs.org/doi/10.1021/acs.chemmater.8b03976; https://www.sciencedirect.com/science/article/abs/pii/S0376738820315672].

5. An attribution of amorphous halo in XRD (at ~20) to diffraction from MXene flakes seems erroneous as neither of Mxene reflections can be indexed in the presented diffraction patterns.

6. The curing agent was not specified in the experimental section.

7. Numerous redundant hypens apper all over the text (ov-en, com-parison, etc). Several hardly-read long sentences appear in the text

Reviewer 3 Report

The authors investigated triboelectric nanogenerators for the health monitoring of human feet, which will be very interesting for lots of readers and a good candidate for health monitoring systems. The manuscript gives an overview of the methodology, modeling, and analytical solution for the nanogenerators, and the paper is well organized and written. My recommendation would be to accept the paper with further revisions as I have the following concerns:

1.      Authors should also note that other energy generators with (especially piezoelectricity) which are being developed all over the world which also have the potential to convert their mechanical energy into electrical energy. This then brings forth the following question: how is energy-harnessing described in this study superior to other energy harvesters-harnessed energy? Please answer this question and mention it in the introduction part while providing a table or explaining it with other technologies (piezoelectric, triboelectric, etc.).

2.      Authors already mentioned various materials of nanogenerator systems, but it is not clear. It would be better if the authors give various triboelectric materials and applications of energy harvesting areas to help the reader focus on the very different experiments whose results are reported. Add also relevant references on vibration/human motion and PDMS-based piezoelectric/triboelectric energy harvesting, like the following:

Advanced Energy Materials 2018, 8, 1702736

Advanced Materials 2018, 30, 1705714

Applied Sciences 2018, 8(2), 213

ACS Applied Materials & Interfaces 2019, 11, 16006-16017

3.      Add relevant references for the main Equation that the authors use here unless the authors found these equations.

4.      Please add the unit in Figure 2(c) for the comsol simulation results.

5.      Based on the results (Figure 3), the authors must explain why 10 wt% of MXene and 0.6 wt% of graphene have a higher output than other ratios. Why do they have this trend?

6.      I believe the test results based on the RH% are related to Figure 3(i), but it’s pointing Figure 3(g) out on line 261. Please fix it and also check other typos in the manuscript. There are so many typos in the manuscript.

7.      It would be great if authors can have COMSOL potential simulation results for the human feet monitoring systems to know why the outer sensor generates higher output.

8.  I would like to see the real demonstration video for the sensor test. Please provide it as supporting information.

Round 2

Reviewer 2 Report

The authors have provided a comprehensive reply to all of my concerns, providing both the reqired data and adequate discussion. Thus, I suggest, the paper can be published in Nanomaterials without further revision. Two minor points seem still be necessary to be paid attention by the authors:

11. (according to comment 2) The size of flakes for MXene and Graphene have to be preferably declared in the text numerically. This seems especially important as the size of flakes should have a strong effect on TENG conductivity and performance.

22. (according to comment 3) I do suggest a huge difference in the optimal graphene and Mxene content in TENG is governed by the distribution of the flakes in the active layer. Even with the images provides in S3 (revised version), one can resolve much rare MXene particles distribution within the layer, despite of much greater content in the film. Chemical interaction of flakes with PDMS appear important. I suggest the authors to comment on the issue and consider it in their further works. Nevertheless, the data provided in S3 seems enough for the readers interested in deeper understanding.

I support the publication after minor revision.

Reviewer 3 Report

The authors have addressed based on my comments, but some answers are still questionable (e.g., based on previous comment #1 & 8), and it brings several questions.

(1) Authors answered my previous comment#1 that TENG is able to achieve higher voltage output than the PG, and TENG is also able to operate in ultra-low frequency (<10 Hz or 1 Hz) to harvest energy in a considerable low cost and flexibility, in which the PG currently has difficulty.

However, PG is also easy to generate output under the frequency level authors mentioned (<10 Hz or 1Hz). Also, PG has flexibility, portability and significant verstility. Hence, I'm asking you this question again. "How is energy-harnessing described in this study superior to other energy harvesters-harnessed energy?"

Also, triboelectric based energy harvesting system has disadvantage, espeically operating under humid atmosphere. How this TENG is better than PG? 

(2) If authors did not record any videos for the feet sensor test, then please record it. It's good time to check the reliability and repeatability. As authors already added vidoes for the LED and electronic watch (additionally), then why not the feet sensor test?

(3) Also add reliability and repeatability data in the main manuscript.  The current data (figure 3) in the manuscript is based on around 5 seconds. Also, the time period for the figure 5 is around 15s. This is not enough to show the reliability and repeatability. 

(4) Based on the Videos 1&2 authors provded, I can only see the results (blinking LED light & turning on the watch). Show the TENG systems and the results together. How did authors generate the power to operate LED and watch? Is this just comming from the power supply? Or from the real TENG authors proposed?

(5) The triboelectric-based nanogenerator system is highly depending on the humidity condition. What was the humidity when the authors measured the generated data? Add it into the figures 3, 4 & 5. 

(6) Have the authors tested the TENGs under different humidity conditions? Add the results based on the different conditions.

(7) What was the humidity condition when authors simulated figure 2 (COMSOL)?

Round 3

Reviewer 3 Report

The authors have addressed well based on my comments. I have no further questions.